# The Role of Mitochondrial Genome Stability and Metabolic Plasticity in Thyroid Cancer

**DOI:** 10.3390/biomedicines13112599

**Published:** 2025-10-23

**Authors:** Lingyu Ren, Wei Liu, Jiaojiao Zheng, Qiao Wu, Zhilong Ai

**Affiliations:** 1Department of Thyroid & Breast Surgery, Zhongshan Hospital, Fudan University, Shanghai 200433, China; 22301050212@m.fudan.edu.cn (L.R.); liu.wei6@zs-hospital.sh.cn (W.L.); zheng.jiaojiao@zs-hospital.sh.cn (J.Z.); 2School of Basic Medical Sciences, Fudan University, Shanghai 200433, China

**Keywords:** thyroid cancer, mitochondrial DNA, metabolic plasticity, mtDNA mutation

## Abstract

Mitochondria play a central role in energy metabolism, redox homeostasis, and signal transduction in the thyroid cells. Increasing evidence indicates that mitochondrial DNA (mtDNA) mutations, copy number variations, and haplogroup-specific polymorphisms are closely associated with metabolic reprogramming and malignant progression of thyroid cancer. This review summarizes recent advances in the understanding of the impact of mitochondrial genome instability and metabolic plasticity on thyroid tumorigenesis. We discuss how mtDNA alterations disrupt oxidative phosphorylation (OXPHOS), trigger adaptive metabolic rewiring, and interact with key oncogenic pathways, such as HIF-1α, BRAFV600E mutations, and TSHR signaling in thyroid cancer. We also highlight the emerging diagnostic and therapeutic potential of mtDNA in thyroid cancer and outline current challenges and future research directions. Gaining deeper insights into the mitochondria–metabolism axis may provide novel biomarkers and metabolic intervention strategies for precision medicine in thyroid oncology.

## 1. Introduction

Thyroid cancer is one of the most common malignancies of the endocrine system, with a notable increase in its incidence in recent decades [1]. From 1992 to 2022, its incidence increased by approximately 150% [2]. Multiple factors contribute to thyroid carcinogenesis, including obesity, smoking, alcohol consumption, and exposure to high-dose radiation. The average age at diagnosis is approximately 50 years, with a significantly higher incidence in females than in males [3]. Histologically, thyroid cancer is classified as papillary thyroid carcinoma (PTC), follicular thyroid carcinoma (FTC), medullary thyroid carcinoma (MTC), and anaplastic thyroid carcinoma (ATC) [4]. Although most thyroid cancers exhibit indolent growth, prognosis varies substantially and is influenced by factors such as patient age, sex, tumor size, and distant metastasis [3]. Therefore, precise delineation of the genetic landscape of thyroid cancer and the identification of key driver mutations and potential therapeutic targets are critical for achieving accurate classification and personalized treatment.

Mitochondria play essential roles in thyroid cells by underpinning thyroid hormone synthesis, energy supply, and cellular signaling regulation. Mitochondria provide reactive oxygen species (ROS) and adenosine triphosphate (ATP) required by thyroid peroxidase (TPO) for hormone biosynthesis [5,6]. In addition, the mitochondria regulate calcium signaling, which participates in hormone secretion. Finally, thyroid follicular epithelial cells exhibit prominent oxidative phosphorylation (OXPHOS) metabolic characteristics that meet their high metabolic demands [7].

In the tumorigenic context, mitochondrial function and genomic architecture undergo significant remodeling. Mitochondrial DNA (mtDNA), a small circular genome independent of nuclear DNA, exhibits a markedly higher mutation rate due to lack of histone protection and exposure to oxidative stress, and is characterized by multi-copy presence, heteroplasmy, and maternal inheritance [8]. Studies have demonstrated that thyroid cancer tissues frequently harbor mtDNA point mutations, copy number alterations, and structural rearrangements [9]. These changes may be closely associated with dysregulated tumor energy metabolism, aberrant signaling pathways, and resistance to apoptosis.

Metabolic plasticity enables tumor cells to adapt to mitochondrial dysfunction, microenvironmental changes, and therapeutic pressure [10]. When OXPHOS is impaired by mtDNA mutations or copy number variations, thyroid cancer cells can flexibly switch between glycolysis, fatty acid oxidation, glutamine metabolism, and other energy pathways to sustain survival and proliferation [11]. This process is tightly regulated by multiple signaling pathways, including HIF-1α [12], the *BRAF*^V600E^ mutation [13], and TSH receptor (TSHR)-related pathways [14]. Metabolic plasticity accounts for the molecular heterogeneity of thyroid cancer and elucidates mechanisms of drug resistance, thereby offering valuable clues for identifying novel therapeutic targets.

## 2. Mitochondrial Genome Instability and Malignant Progression of Thyroid Cancer

### 2.1. Molecular Characteristics of mtDNA

mtDNA is a double-stranded circular molecule approximately 16.5 kb in length that encodes 37 genes, including 13 respiratory chain protein subunits, 22 transfer RNAs (tRNAs), and 2 ribosomal RNAs (rRNAs) [15]. Due to the high levels of ROS within mitochondria, combined with the lack of histone protection and limited DNA repair mechanisms, mtDNA is prone to point mutations, deletions, insertions, and copy number variations [16], with a mutation rate estimated to be 10 to 100 times higher than that of the nuclear genome [17]. This high mutability provides a potential foundation for metabolic reprogramming and malignant progression for tumors.

### 2.2. mtDNA Point Mutations in Thyroid Cancer

mtDNA point mutations are widespread across various cancers, including renal carcinoma, colorectal cancer [18], and thyroid cancer, particularly in oncocytic thyroid tumors (Table 1) [19]. These tumors are characterized by an abundance of mitochondria, typically reflecting compensatory mitochondrial proliferation triggered by impaired electron transport chain function [20]. Disruptive mutations in mtDNA-encoded complex I (CI) subunits have been established as a hallmark molecular feature of oncocytic thyroid carcinoma (Figure 1) [21,22], showing strong statistical association with the oncocytic phenotype. The loss of these mutations in primary cell culture suggests their in vivo maintenance depends on specific selective pressures, such as hypoxia, implying the mutation requires this synergistic interplay to exert its full effect [21]. Similarly, the elevated D-loop mutation rate observed in oncocytic tumors is best interpreted as a byproduct of compensatory mtDNA replication, which is itself a response to the characteristic mtDNA deletions, rather than an underlying driver of tumorigenesis [23,24].

Beyond oncocytic thyroid tumors, other histological types such as PTC and FTC also exhibit increased mitochondrial content. Immunohistochemical analyses have confirmed a significant increase in mitochondrial mass in the majority of PTC samples compared to adjacent normal tissue [25]. At the genetic level, pathogenic frameshift mutations in genes encoding complex I subunits have been identified in a subset of PTCs, with a notable tendency toward homoplasmy [25,26]. Importantly, germline sequence variants affecting complex I and IV have been reported at higher frequencies in the non-tumorous thyroid parenchyma adjacent to malignant lesions [27]. This finding suggests that an inherited predisposition, reflected in specific mtDNA polymorphisms, may create a permissive background upon which somatic mutations accumulate during the evolution of thyroid carcinomas, thereby positioning mtDNA alterations as potential co-factors in thyroid tumorigenesis beyond the oncocytic subtype.

Regarding the noncoding control region, studies have reported divergent findings on the mutation frequency in the D-loop of PTCs. While some reports describe a markedly increased mutation burden in this region [28], others observed a lower mutation rate in the D310 mononucleotide repeat compared to other epithelial malignancies, with no clear correlation to tumor grade [29]. This discrepancy may reflect methodological differences, sample heterogeneity, or varying interpretations of the biological role of D-loop mutations. The latter view posits that these alterations may arise as epiphenomena of clonal mitochondrial expansion rather than as functional drivers of tumor progression, consistent with observations in oncocytic tumors [23,24,29]. Additionally, somatic instability in mitochondrial microsatellite regions—such as the T16189C variant and dinucleotide repeats—has been found to occur more frequently in PTC than in benign nodules, pointing to a possible role for mtDNA replication infidelity in malignant transformation [30].

Of note, the tall cell variant (TCV) of PTC shares key features with oncocytic tumors [31,32]. Studies indicate that TCV of PTC exhibits more prominent mitochondrial accumulation, higher mtDNA mutation frequencies, and complete loss of CI subunit NDUFS4 compared with other PTC subtypes, proposing that homoplasmic or highly heteroplasmic mtDNA mutations may serve as defining molecular features of this aggressive histological variant [31].

Nonetheless, validating the pathogenicity of mtDNA mutations remains challenging, primarily due to heteroplasmy dynamics and the difficulty in distinguishing driver from passenger events. mtDNA heteroplasmy, the coexistence of mutant and wild-type genomes, requires the mutation load to exceed a certain threshold (typically >60%) to manifest phenotypically [33]. The heteroplasmy status of D-loop variants observed in thyroid lesions is conceptually consistent with this threshold model, demonstrating a full spectrum from low heteroplasmy to homoplasmy [34]. However, as most D-loop mutations are of uncertain pathogenic significance [34,35], their phenotypic impact—if any—would be expected to correlate with their mutation load. In addition, the occurrence of identical D-loop mutations in both benign and malignant lesions implies that many of these alterations may function as passenger events, secondary to tumor development rather than primary oncogenic drivers [8,34]. Thus, definitive establishment of mtDNA mutations as pathogenic in thyroid cancer requires functional validation to clarify their causal contributions, distinct from the backdrop of clonal expansion and nuclear genomic instability.

**Table 1 biomedicines-13-02599-t001:** Alterations in mtDNA in thyroid cancer.

Gene/Region	Variation	Type	Amino Acid (aa) Change	Cancer Type	Refs.
ND1/CI	3308 T→C	Transition	M1T	PTC	[23]
	4216 T→C	Transition	Y303H	PTC, IC, FA	[23]
	4225 A→G	Transition	M306V	PTC	[23]
	4248 T→C	Transition	-	PTC	[23]
	3842 G→A	Transition	W179X	FTC	[21,36]
	3910 G→A	Transition	E202K	FTC	[23]
	3594 C→T	Transition	-	PTC	[27]
	3526 G→A	Transition	A74T	PTC	[27]
	3571 ins C	Insertion	Frameshift	PTC-TCV	[31]
	3955 G→C	Transversion	A217P	PTC-TCV	[31]
	3380 G→A	Transition	R25Q	PTC-TCV	[31]
ND2/CII	4917 A→G	Transition	N149D	PTC, IC, FA	[23]
	4883 C→T	Transition	-	PTC	[23]
	5298 A→G	Transition	I-V	PTC	[26]
	5408 del A	Deletion	Frameshift	PTC	[23]
	4940 C→T	Transition	-	PTC	[27]
	4611–4612 del A	Deletion	Frameshift	PTC	[25]
	4605 del A	Deletion	Frameshift	PTC-TCV	[31]
	4611 del A	Deletion	Frameshift	PTC-TCV	[31]
ND3/CI	10,398 A→G	Transition	T113A	PTC, FTC, HT	[23]
	10,116 del AT	Deletion	31X	PTC	[21]
	10,320 G→A	Transition	V88I	PTC	[27]
ND4/CI	11,812 A→G	Transition	-	PTC, IC, FA	[23]
	11,126 G→A	Transition	E-K	PTC	[26]
	11,736 T→C	Transition	L326P	PTC	[21]
	11,840 C→T	Transition	-	PTC	[27]
	11,179–11,180 ins T	Insertion	Frameshift	PTC	[25]
	11,873 ins C	Insertion	Frameshift	PTC-TCV	[31]
	11,038 del A	Deletion	Frameshift	PTC-TCV	[31]
	11,364 C→T	Transition	A202V	PTC-TCV	[31]
	10,946 ins C	Insertion	Frameshift	PTC-TCV	[31]
	11,475 G→A	Transition	G239D	PTC-TCV	[31]
ND4L/CI	10,691 C→G	Transversion	-	FTC	[27]
ND5/CV	13,617 T→C	Transition	-	PTC	[23]
	13,514 A→G	Transition	D-G	PTC	[26]
	13,943 C→T	Transition	T536M	PTC	[27]
	12,967 A→C	Transversion	T211P	FTC	[27]
	13,805 C→T	Transition	A490V	PTC-TCV	[31]
ND6	14,512 T→C	Transition	L121P	HT	[23]
	14,417 A→G	Transition	V-A	PTC	[26]
	14,451–14,452 ins T	Insertion	Frameshift	PTC	[25]
	14,660–14,661 del	Deletion	Frameshift	PTC-TCV	[31]
	14,584 del T	Deletion	Frameshift	PTC-TCV	[31]
CYTB/CIII	15,326 A→G	Transition	T193A	PTC, FTC	[23]
	15,179 G→A	Transition	V144M	PTC	[23]
	15,301 G→A	Transition	-	FTC	[23]
	15,262 T→C	Transition	-	PTC	[23]
	15,280 C→T	Transition	-	PTC	[27]
	14,864 T→A	Transversion	C40S	PTC-TCV	[31]
COI/CIV	7389 T→C	Transition	Y495H	PTC	[23]
	7444 G→A	Transition	Ter496K	PTC	[23]
	7424 A→G	Transition	-	FTC	[23]
	7441 C→A	Transversion	S513Y	PTC	[21]
	5979 G→A	Transition	A26T	PTC-TCV	[31]
COII/CpIV	7705 T→C	Transition	-	PTC	[23]
	8251 G→A	Transition	-	PTC	[23]
	7785 T→ C	Transition	I67T	PTC	[27]
	7658 G→A	Transition	D25N	PTC-TCV	[31]
COIII/CpIV	9380 G→A	Transition	-	HT	[23]
	9755 G→A	Transition	-	PTC	[23]
	9932 G→A	Transition	-	PTC	[23]
	9899 T→C	Transition	-	PTC	[23]
	9948 G→A	Transition	V-I	PTC	[26]
ATPase 6/CpV	8725 A→G	Transition	T67A	PTC	[21]
ATPase 8/CpV	8414 C→T	Transition	L16F	PTC	[23]
12S rRNA	663 A→G	Transition		PTC	[23]
	709 G→A	Transition		PTC, IC, FA	[23]
	710 T→C	Transition		PTC	[23]
16S rRNA	3197 T→C	Transition		PTC	[23]
tRNA Asp	7521 G→A	Transition		PTC, IC	[23]
tRNA Arg	10,463 T→C	Transition		PTC, IC, FA	[23]
tRNA Leu1	3244 G→A	Transition		PTC-TCV	[31]
tRNA Leu2	12,308 A→G	Transition		PTC	[23]
tRNA Ser	7476 C→T	Transition		PTC	[26]

FA, follicular adenoma; FTC, follicular thyroid carcinoma; HT, Hashimoto’s thyroiditis; IC, insular carcinoma; PTC, papillary thyroid carcinoma; PTC-TCV, papillary thyroid carcinoma tall cell variant.

### 2.3. Association of Mitochondrial Haplogroups with Clinical Phenotypes

Mitochondrial haplogroups, defined by specific single-nucleotide polymorphisms (SNPs) within mtDNA, represent maternally inherited lineages with population specificity and interethnic differences (Figure 1). Previous research has linked distinct haplogroups to disease susceptibility and prognosis, such as haplogroup D5 with breast cancer risk [37], and haplogroup U with prostate and colorectal cancers [38].

The associations between mitochondrial haplogroups and thyroid cancer risk are varied across studies, collectively underscoring the complexity of their contributions. For instance, an elevated risk of thyroid cancer has been associated with haplogroup D4a in certain Chinese cohorts [39], whereas haplogroup K has been correlated with a potential protective effect in other populations [40]. Similarly, haplogroup U has been significantly linked to both benign and malignant thyroid tumors, in contrast to haplogroup J, which may be protective in some contexts [30].

These findings imply that mitochondrial haplogroups may have ethnicity, tissue, and disease-stage-specific roles in thyroid carcinogenesis [41]. Nevertheless, the precise molecular mechanisms—including the potential interplay between haplogroup-defining polymorphisms, somatic mtDNA mutations, and the nuclear genetic background—remain largely unresolved. Future studies integrating comprehensive population genetics with rigorous functional assays are required to clarify these interactions and define the pathogenic relevance of mitochondrial haplogroups in thyroid cancer.

### 2.4. mtDNA Copy Number Variations and Tumor Behavior

Alterations in mtDNA copy number are important markers of mitochondrial functional adaptation. While elevated mtDNA copy number has been consistently associated with tumorigenesis and poor prognosis in several malignancies—including lung [42], breast [43], liver [44], and gastric cancers [45]—its role in thyroid cancer remains contentious and analytically nuanced.

In PTC, tissue-based studies have reported a nearly fourfold increase in mtDNA copy number compared to adjacent normal tissue [9], with some analyses suggesting that high mtDNA content may double the risk of developing PTC (Figure 1) [46]. However, these tissue-based observations stand in stark contrast to findings from blood-based assays (Figure 1). For instance, reduced mtDNA copy number has been reported in the peripheral blood of patients with malignant thyroid nodules relative to those with benign nodules [47]. Further complicating the picture, a 2023 large-scale genetic study involving over 72,000 individuals found no significant causal relationship between mtDNA copy number and thyroid cancer risk [48].

These discrepant findings reflect several underlying methodological and biological challenges. The observed inconsistencies may be attributed to tumor subtype heterogeneity, differences in sample source (tissue vs. blood), and variations in detection methodologies. Blood-based mtDNA copy number measurements may not accurately reflect tumor mitochondrial content, as the mechanisms governing mtDNA release into circulation—and its possible origins from immune or other non-malignant cells [49]—remain poorly understood.

Future studies should aim to clarify the tissue-specific biology of mtDNA replication in thyroid cancer. Given the central role of the organ in systemic metabolism, the local hormonal milieu—particularly thyroid hormone signaling [50]—along with oxidative stress responses and nuclear-mitochondrial cross-talk, constitutes a critical axis for investigation. Well-designed, multicenter investigations using standardized detection methods and matched tissue-blood samples are needed to determine whether mtDNA copy number has true discriminative or prognostic value in thyroid cancer management.

## 3. Role of Mitochondrial Metabolic Plasticity in Thyroid Cancer

### 3.1. Core Features and Manifestations of Metabolic Plasticity

Metabolic plasticity represents a critical survival strategy by which tumor cells adapt to microenvironmental stresses such as hypoxia, nutrient deprivation, and therapeutic interventions. Its hallmark is the dynamic regulation of major metabolic pathways, including glycolysis, OXPHOS, fatty acid metabolism, and amino acid metabolism, to maintain energy supply and biosynthetic balance [11].

Thyroid cancer cells commonly exhibit pronounced metabolic reprogramming [51]. For example, the Warburg effect, defined as preferential glycolysis under aerobic conditions [52], has been repeatedly observed in PTC. Studies have demonstrated that the expression of glucose transporters (GLUTs) negatively correlates with tumor differentiation, while glycolytic activity positively correlates, suggesting that upregulation of GLUT enhances glucose uptake to compensate for the relatively low ATP yield of glycolysis [53]. Key glycolytic and ancillary enzymes, including G6PD, PGK1, LDHA, and PHGDH, are frequently upregulated in PTC tissues, highlighting enhanced glucose utilization [54]. Beyond glycolysis, both fatty acid synthesis and oxidation are elevated, providing additional energy and building blocks to sustain tumor growth [11].

The tricarboxylic acid (TCA) cycle and glutamine metabolism further contribute to tumor proliferation [55,56,57]. Accumulation of TCA intermediates such as pyruvate and fumarate in PTC, accompanied by increased glutamine and glutamate levels, supplies substrates for nucleotide and lipid biosynthesis [58]. Metabolic profiling indicates stage-specific reprogramming, differentiated thyroid cancers (e.g., PTC-B and FTC-R) show mutation-associated metabolic signatures (*BRAF*, *RAS*), whereas ATC favors one-carbon and pyrimidine metabolism, reflecting shifts that support rapid growth and aggressiveness [59].

Collectively, thyroid cancer cells maintain survival and invasiveness in hostile environments through “multi-pathway coexistence and the dynamic switching” of metabolic pathways. This process opens new avenues for metabolism-targeted therapies.

### 3.2. Regulatory Mechanisms of Metabolic Plasticity

Metabolic plasticity depends on the coordinated regulation of various transcription factors and signaling pathways. In thyroid cancer, HIF-1α, *BRAF* mutation, and TSHR signaling are considered key regulators (Figure 2).

HIF-1α is the master transcription factor responding to hypoxia, and its expression stabilizes under OXPHOS impairment or hypoxic conditions. It upregulates GLUT and multiple glycolytic enzymes (HK2, PFK1, PKM2, and LDHA) [11], and amplifies the Warburg effect [60]. In PTC, hypoxia has been associated with co-upregulation of HIF-1α, YAP, and GLUT1, enhancing glucose uptake and lactate production through the HIF-1α/YAP axis [61]. In addition, HIF-1α activation can induce TERT expression, which in turn modulates the mTOR pathway and stimulates autophagy, forming a regulatory axis that supports tumor proliferation, migration, and invasion [62].

*BRAF*^V600E^ mutation is the most common driver mutation in PTC [63]. Studies have found that *BRAF*-mutant thyroid cancers exhibit enhanced metabolic reprogramming and aggressive characteristics [64]. It enhances glycolytic capacity by activating the MAPK–ERK–DRP1 pathway to increase HK2 expression [65] and has also been associated with elevated PKM2 expression, suggesting a role in reinforcing glycolytic flux and metabolic remodeling [66].

TSHR, a thyroid cell-specific marker, is also implicated in the metabolic regulation of thyroid cancer. Within the tumor microenvironment, monocyte-derived dendritic cells (moDCs) can secrete TSH, which promotes the proliferation and invasion of TSHR-high tumor cells through the TSHR–adenylyl cyclase (AC)–protein kinase A (PKA)–JNK signaling axis [67]. In addition, TSH/TSHR signaling has been shown to suppress fatty acid synthase (FASN) expression in adipocytes via PKA- and ERK-dependent mechanisms, thereby linking TSH action to lipid metabolism [68]. TSH also regulates JNK activity in human thyroid cells in a concentration-dependent manner, with low levels stimulating and high levels inhibiting JNK through cross-talk between Gi/PKC and cAMP/PKA pathways [69]. Together, these findings highlight the multifaceted role of TSHR signaling in coordinating tumor cell proliferation, invasion, and metabolic adaptation.

In summary, HIF-1α, *BRAF* mutations, and TSHR signaling synergistically contribute to metabolic plasticity in thyroid cancer, representing key mechanisms underlying metabolic reprogramming and drug resistance. Additional pathways, such as PI3K/Akt and AKT/mTOR/HK2 axis, may also be involved and warrant further investigation.

## 4. Interaction Mechanisms Between Mitochondrial Genome Variations and Thyroid Tumor Metabolic Plasticity

Mitochondrial genome variations, including point mutations, haplogroup differences, and copy number alterations, not only serve as key drivers of metabolic reprogramming in thyroid cancer cells but are also subject to feedback regulation by the metabolic state (Figure 3). The bidirectional interplay between these factors collectively shapes metabolic heterogeneity and adaptability of tumors.

### 4.1. Interaction Between mtDNA Point Mutations and Metabolic Remodeling

mtDNA point mutations have been shown to induce electron transport chain (ETC) dysfunction in various cancers, thereby driving metabolic remodeling. Emerging evidence indicates that such mutations play a role in metabolic reprogramming in colorectal cancer [70] and during leukemia development [71]. Studies using the XTC.UC1 oncocytic thyroid carcinoma cell line demonstrated that mtDNA mutations alone can lead to respiratory chain defects [72]. In cybrid models, specific variants—such as a frameshift in ND1 and a missense mutation (E271K) in CYTB—were found to impair complex I and III function, reduce ATP production, and increase ROS, confirming their pathogenic role in metabolic remodeling [72]. Consistent with these observations, ETC damage has been detected in a large proportion of PTC samples, with mutations in complex I subunits being particularly prominent [25]. Moreover, heteroplasmic ND5 mutation altered ROS generation and apoptosis that favored tumorigenesis, whereas the same mutation in a homoplasmic state suppressed tumor formation. This study underscores that the phenotypic impact of mtDNA mutations may depend not only on the affected respiratory chain complex but also on heteroplasmy levels. ROS imbalance, in this context, appears to serve as a mechanistic bridge between mitochondrial genetic defects and metabolic remodeling [73]. Beyond variants in the coding region, mutations in the noncoding D-loop are of critical importance. Pan-cancer analyses indicate that somatic D-loop mutations are influenced by tumor-specific evolutionary selection [74]. Functionally, mutations occurring in the more conserved non-hypervariable segments (non-HVS) show a significant association with reduced mtDNA copy number, directly linking them to impaired mitochondrial biogenesis [74]. This notion is further supported by studies demonstrating that chemotherapy-induced D-loop mutations, particularly those in hypervariable segment 1 (HVS1), are associated with decreased mtDNA copy number and increased chemotherapy resistance [75]. Although metabolic pathway alterations were not directly validated in some studies, their findings support the notion that mtDNA mutations can indirectly drive metabolic reprogramming via mitochondrial dysfunction [74,75]. However, there remains no conclusive evidence that mtDNA mutations directly control glycolysis, fatty-acid metabolism, or other remodeling processes through transcriptional, epigenetic, or signaling pathways.

Notably, the metabolic state itself may reciprocally influence mtDNA mutation frequency or selection. Excessive ROS generated by altered metabolism can exacerbate mtDNA damage, potentially establishing a positive feedback loop [76].

Therefore, the causal role of mtDNA mutations in metabolic remodeling remains to be fully elucidated. Existing approaches such as cybrid models provide a means to control for nuclear genetic background [77], yet further mechanistic studies are required. The recent development of mitochondrial-specific editing technologies, including base editors and experimental CRISPR-mt systems, may enable more precise validation of pathogenic effects in tumor models [78,79].

A few recent studies have employed cybrid models to assess the functional impact of mtDNA mutations while controlling for nuclear DNA background. For instance, a study introducing the G3842A mutation identified in thyroid cancer into a cybrid line showed impaired complex I activity, reduced OCR and ATP production, and elevated ROS, suggesting a direct pathogenic effect [36]. Another melanoma cybrid study found that pathogenic mtDNA variants impaired metastatic dissemination, and that during tumor growth there is selection against dysfunctional mtDNA alleles [80]. However, such models remain relatively rare in thyroid cancer. Many existing reports document associations between mtDNA mutations or expression changes and metabolic phenotypes without using matched nuclear backgrounds [9,81]. Therefore, further work using cybrid or equivalent systems in thyroid cancer is needed to clarify causality, threshold effects (heteroplasmy), and the influence of nuclear-mitochondrial interactions.

### 4.2. Mitochondrial Haplogroups and Metabolic Tendencies

Mitochondrial haplogroups serve not only as population genetic markers but also influence individual metabolic traits. Potential mechanisms include modulation of the respiratory chain subunit conformation, ETC assembly efficiency, ROS production levels, and substrate utilization patterns.

Different haplogroups display distinct metabolic phenotypes. For example, haplogroup N9a has been associated with reduced OXPHOS activity, while haplogroup D5 correlated with suppressed mitochondrial respiration and decreased membrane potential in breast cancer and osteosarcoma [37]. Haplogroup M8a has been linked to impaired OXPHOS efficiency, as reflected by reduced mitochondrial light strand transcription, lower ATP production, diminished maximal respiratory capacity, and increased ROS levels [82].

In thyroid cancer, systematic investigations of the associations between haplogroups and metabolic features remain scarce. A potential protective role of haplogroup K has been suggested [40]. This haplogroup has also been implicated in other conditions including Parkinson’s disease [83], autism spectrum disorder [84] and narcolepsy type 1 [85]. And in narcolepsy type 1, haplogroup K was associated with elevated triglyceride levels [85], pointing to a possible role in energy metabolism modulation.

It is worth noting that the metabolic impact of mitochondrial haplogroups is strongly modulated by nuclear genetic background [86], cell type [87], and microenvironmental context [88]. Without controlling for these variables, study outcomes may be biased. Hence, clarifying the role of haplogroups in thyroid cancer metabolic regulation requires large-scale population studies coupled with gene editing validation.

### 4.3. mtDNA Copy Number Variation and Metabolic Adaptation

Alterations in mtDNA copy number represent an adaptive strategy through which tumor cells counteract mitochondrial dysfunction, exerting bidirectional influence on metabolic reprogramming. Changes in copy number have been shown to directly affect tumor metabolic gene expression, particularly in pathways related to the TCA cycle and ETC [89].

An increase in copy number, frequently observed in OXPHOS-dependent tumor subpopulations, enhances respiratory chain protein expression, thereby boosting ATP synthesis and NAD^+^ regeneration [90]. This elevation helps buffer the detrimental effects of mtDNA mutations and promotes metabolic homeostasis. Conversely, a decrease in mtDNA copy number, often triggered by hypoxic conditions or nutrient deprivation, reduces ROS production and diminishes cellular sensitivity to apoptosis [91]. This reduction concomitantly facilitates a shift towards enhanced glycolysis, enabling the tumor to sustain its energy demands under stress [92].

In thyroid cancer, increased mtDNA copy numbers have been detected in both oncocytic and non-oncocytic thyroid cancers and this elevation has been associated with poor prognosis. Their study suggested that copy number elevation appears to be linked with enhanced amino acid metabolism and mitochondrial stress responses, constituting an integral component of metabolic reprogramming [93].

In summary, the regulation of mtDNA copy number represents a critical nexus in the interaction between the mitochondrial genome and tumor metabolism, with its dynamic modulation playing a vital role in sustaining tumor survival and progression.

## 5. Clinical Applications of mtDNA in Thyroid Cancer Diagnosis and Therapy

With a deepening understanding of the role of the mitochondrial genome in tumor biology, the potential clinical applications of mtDNA in thyroid cancer have gradually emerged, encompassing early diagnosis, risk stratification, and targeted therapy. mtDNA mutations, copy number alterations, and their release into body fluids form the basis for developing novel molecular biomarkers [94]. For instance, mutations in the mtDNA D-loop region, a regulatory hotspot, are observed in approximately 23.9% of PTC samples [95]. However, their frequent occurrence in benign thyroid lesions [24] limits diagnostic specificity, underscoring the need for combined biomarker strategies. In addition to sequence alterations, changes in mtDNA copy number have been reported, with significantly higher levels in PTC and adenoma tissues than in nodular goiters, suggesting a potential criterion for distinguishing benign from malignant nodules. Nevertheless, the lack of systematic evaluation of their sensitivity and specificity further constrains their clinical translation.

Circulating cell-free mitochondrial DNA (cf-mtDNA) demonstrates a complex and context-dependent profile in thyroid cancer. In PTC, plasma cf-mtDNA levels are significantly reduced, in contrast to the elevated mtDNA content typically observed within tumor tissues [96,97]. Although this alteration demonstrates high specificity, its clinical utility is limited by poor detection sensitivity [96]. Evidence from MTC, despite primarily focusing on nuclear DNA mutations, highlights a similar challenge—the insufficient sensitivity of current assays to reliably detect tumor-derived DNA in plasma [98]. The biological mechanism underlying the reduced cf-mtDNA in PTC remains unclear, potentially involving impaired mitochondrial release, oxidative stress-induced fragmentation [99], immune-mediated clearance as a damage-associated molecular pattern (DAMP) [100], and preferential packaging into extracellular vesicles [101]. Notably, methodological variability—including differences in sample processing, DNA extraction, and quantification—significantly affects cf-mtDNA measurements, contributing to inconsistent results across studies and limiting clinical applicability [101,102].

Beyond diagnostics, mtDNA-encoded respiratory chain proteins are involved in tumor energy metabolism, and their mutation-induced OXPHOS dysfunction, ROS accumulation, and metabolic remodeling provide a basis for exploring therapeutic approaches [36,41]. Mutations in ND subunits (complex I) and COX1/2 (complex IV) can impair OXPHOS, highlighting potential targets for drug development [103]. Based on these alterations, several mitochondria-targeted strategies have been investigated. Complex I inhibitors (e.g., IACS-010759) [104,105,106] and glutaminase inhibitors (e.g., CB-839) [107,108,109] have shown activity in preclinical studies and early clinical trials. Pharmacological ROS modulators, such as elesclomol, and inhibitors of antioxidant pathways, including NRF2 signaling, have also been examined to modify tumor redox balance [110,111,112,113]. In addition, nanoparticle-based approaches, such as KLA-Au systems, have been reported to selectively accumulate in mitochondria, enhance oxidative damage, and potentially promote mtDNA release and innate immune activation [114,115,116].

Besides exploiting these metabolic and redox vulnerabilities, new techniques are being developed to directly target mitochondrial genomes. Mitochondria-specific nucleases (e.g., mitoTALENs [117] and mitoARCUS [118]) and base editing tools (e.g., DdCBEs [119,120] and high-fidelity variants [121]) have demonstrated the ability to selectively reduce mutant mtDNA or perform precise base conversions in mammalian models [122]. These methods remain at an early stage but illustrate potential directions for addressing pathogenic mtDNA alterations.

In summary, mtDNA represents a potential biomarker and therapeutic target in thyroid cancer, but its clinical translation remains limited (Table 2). Diagnostic applications, including D-loop mutations, mtDNA copy number alterations, and circulating cell-free mtDNA, are constrained by limited specificity, low abundance, and methodological variability, which affect reproducibility and comparability across studies. Therapeutic approaches targeting mitochondrial metabolism, redox balance, nanodrug delivery, or employing genome editing tools show preclinical promise but remain at an early stage, with functional validation and safety evaluation required before clinical implementation. Overall, standardized detection methods, robust functional studies, and prospective multicenter trials are essential to establish the clinical utility of mtDNA in both diagnosis and treatment of thyroid cancer.

## 6. Conclusions and Future Perspectives

The dynamic interplay between mitochondrial genome stability and metabolic plasticity constitutes the fundamental basis for thyroid cancer initiation and progression. A thorough understanding of the causal relationships between mtDNA variations and energy metabolism may provide novel insights for early screening, risk stratification, and targeted intervention. Overcoming current technical and mechanistic barriers is imperative to shift from descriptive metabolic observations to actionable metabolic intervention strategies, thereby advancing the precision prevention and treatment of thyroid cancer into a new era.

## Figures and Tables

**Figure 1 biomedicines-13-02599-f001:**
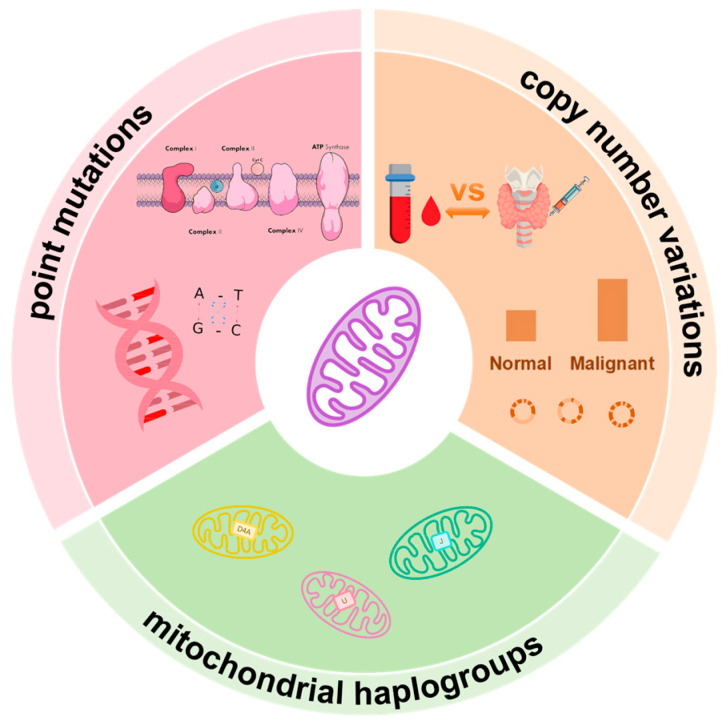
Major types of mitochondrial DNA (mtDNA) alterations in thyroid cancer. Mitochondrial genome instability manifests in three major forms: point mutations, nucleotide substitutions affecting coding or regulatory regions; haplogroups, population-specific variants with potential implications for disease susceptibility; and copy number variations, changes in mtDNA content reflecting mitochondrial biogenesis and dysfunction.

**Figure 2 biomedicines-13-02599-f002:**
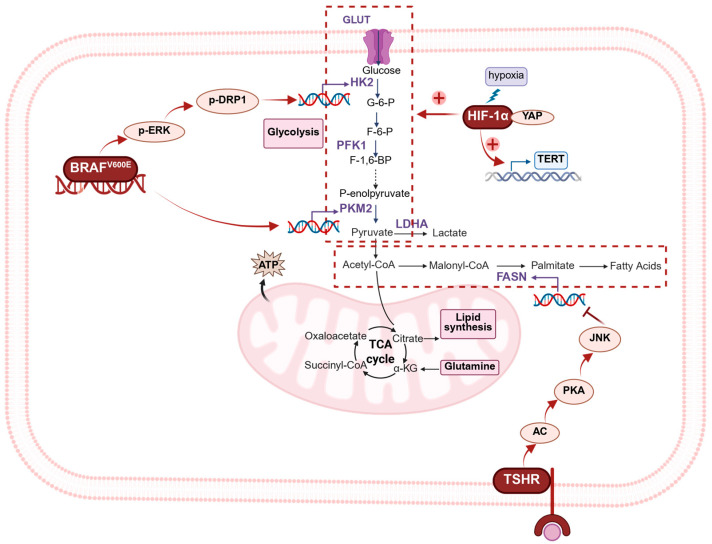
Key regulators of metabolic plasticity in thyroid cancer. HIF-1α, *BRAFV600E*, and TSHR signaling drive glycolysis, TCA cycle adaptation, and lipid synthesis, converging to establish metabolic plasticity and promote tumor progression.

**Figure 3 biomedicines-13-02599-f003:**
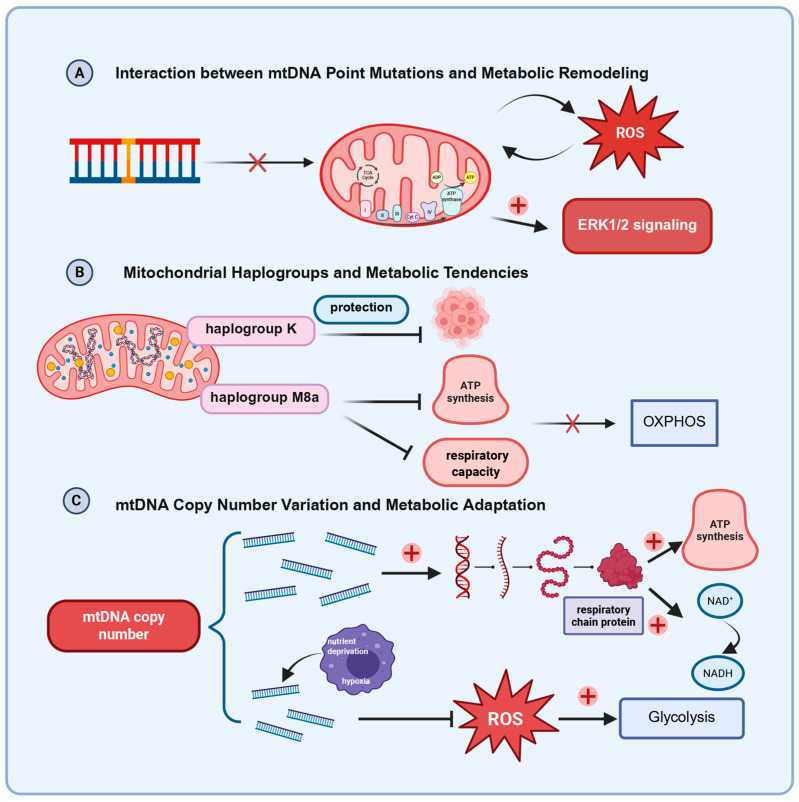
Mitochondrial genome variations and metabolic plasticity in thyroid cancer. (**A**) mtDNA point mutations disrupt respiratory chain function, elevating ROS and activating oncogenic signaling; (**B**) Mitochondrial haplogroups modulate OXPHOS efficiency and metabolic tendencies; (**C**) Dynamic mtDNA copy number changes enable tumor cells to balance ROS control with shifts between OXPHOS and glycolysis. Together, these mechanisms shape metabolic heterogeneity and adaptability in thyroid cancer.

**Table 2 biomedicines-13-02599-t002:** Clinical Applications of mtDNA in Thyroid Cancer Diagnosis and Therapy.

Category	Alteration/Target	Mechanism/Observation	Potential Clinical Application	Limitations	Refs
Diagnostic	mtDNA D-loop mutations	Regulatory hotspot; frequent in PTC but also in benign lesions	Early detection, risk stratification	Limited specificity; occurs in benign lesions; heteroplasmy complicates interpretation	[24,95]
	mtDNA copy number alterations	Increased in PTC and adenomas vs. nodular goiter	Potential biomarker to distinguish benign vs. malignant nodules	Conflicting results between tissue and blood; methodological variability; lack of standard thresholds	[46]
	Circulating cf-mtDNA	Reduced in PTC plasma	Minimally invasive detection; prognostic assessment	Low abundance; detection sensitivity limited; blood origin may include non-tumor cells; inter-study variability	[96,98,99,100,101,102]
Therapeutic—Metabolic Targeting	Complex I/ND subunits, COX1/2 mutations	OXPHOS dysfunction, ROS accumulation	Complex I inhibitors (IACS-010759), glutaminase inhibitors (CB-839)	Mostly preclinical; unclear efficacy and safety in patients; heterogeneity in tumor metabolic profiles	[103,104,105,106,107,108,109]
Therapeutic—Redox Modulation	ROS imbalance/antioxidant pathways	Excessive ROS, metabolic remodeling	Elesclomol, NRF2 inhibitors	Early stage; systemic toxicity possible; context-dependent efficacy	[110,111,112,113]
Therapeutic—Nanoparticles	KLA-Au systems	Mitochondrial accumulation, oxidative damage, mtDNA release	Induction of innate immunity, potential anti-tumor effect	Preclinical; pharmacokinetics, targeting efficiency, and safety not fully established	[114,115,116]
Therapeutic—Genetic Editing	mtDNA mutations	Targeted mutation reduction or base editing	mitoTALENs, mitoARCUS, DdCBEs	Early stage; delivery efficiency, off-target effects, and nuclear-mitochondrial interaction concerns	[117,118,119,120,121,122]

## Data Availability

Not applicable.

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
