# Peer review of "The Role of Mitochondrial Genome Stability and Metabolic Plasticity in Thyroid Cancer"

_biomedicines, 2025, doi:10.3390/biomedicines13112599_

Round 1
Reviewer 1 Report
Comments and Suggestions for Authors
This manuscript has undergone a comprehensive review and is found to require minor revisions before it can be accepted for publication. The following requests for substantial modifications need to be addressed to enhance the manuscript's quality and suitability for final acceptance:
1) The statement that mtDNA has a 10–100 times higher mutation rate than nuclear DNA is repeated in both the introduction (Page 2, line 51) and section 2.1 (Page 2, line 75). It would be clearer to mention it only once.
2) The statement ‘From 1992 to 2018, its incidence increased by more than 200%’ needs a supporting reference or accuracy check, since the current citation [1] may not directly confirm this figure.
3) Typographical: On page 6, line 265, ‘my reciprocally’ should be corrected to ‘may reciprocally’.
4) The clinical applications section could be expanded by adding a discussion on mitochondria-targeted therapies, including metabolic inhibitors, ROS modulators, and new mitochondrial editing approaches. Covering these points would give readers a clearer and more complete view of how mitochondrial research may translate into thyroid cancer treatment.
Comments on the Quality of English LanguageThe manuscript contains several grammatical and spelling issues that need to be addressed for clarity and accuracy. It may require professional English editing to ensure precision and readability.
Author Response
Dear reviewer,
We are thankful for the positive responses. Following the suggestions, we thoroughly revised the manuscript and addressed the issues you mentioned. Revisions are marked using the “Track Changes” function. The corrections in the paper and the response are as follows:
Comments 1. The statement that mtDNA has a 10–100 times higher mutation rate than nuclear DNA is repeated in both the introduction (Page 2, line 51) and section 2.1 (Page 2, line 75). It would be clearer to mention it only once.
Response: Thank you for this suggestion. To avoid redundancy, we have removed the statement regarding the “10–100 times higher mutation rate of mtDNA” from the Introduction, as you recommended. The point is now retained only in Section 2.1, where it is most contextually appropriate.
Comments 2. The statement ‘From 1992 to 2018, its incidence increased by more than 200%’ needs a supporting reference or accuracy check, since the current citation [1] may not directly confirm this figure.
Response: Thank you for this critical observation. We have updated our statement to: "From 1992 to 2022, its incidence increased by approximately 150%" based on the following reference.
[2] National Cancer Institute. Cancer Stat Facts: thyroid cancer. 2025 [cited 2025 10-02]; Available from: https://seer.cancer.gov/statfacts/html/thyro.html.
Comments 3. Typographical: On page 6, line 265, ‘my reciprocally’ should be corrected to ‘may reciprocally’.
Response: Thank the you for pointing out this typographical error. The correction has been made as suggested. "my reciprocally" has been changed to "may reciprocally" on page 6, line 275.
Comments 4. The clinical applications section could be expanded by adding a discussion on mitochondria-targeted therapies, including metabolic inhibitors, ROS modulators, and new mitochondrial editing approaches. Covering these points would give readers a clearer and more complete view of how mitochondrial research may translate into thyroid cancer treatment.
Response: Thank you for this valuable suggestion. In the revised manuscript, we have expanded the clinical applications section to include a discussion of mitochondria-targeted therapeutic strategies in thyroid cancer. Specifically, we now cover: (1) metabolic inhibitors, such as complex I inhibitors (IACS-010759) and glutaminase inhibitors (CB-839), (2) pharmacological ROS modulators and inhibitors of antioxidant pathways (including NRF2 signaling), (3) nanoparticle-based approaches that selectively target mitochondria and modulate oxidative stress, and (4) emerging mitochondrial genome editing tools, including mitoTALENs, mitoARCUS, and DdCBE-based base editors. References have been added to support each strategy ([86–105]). These additions provide a more comprehensive view of how mitochondrial research may be translated into therapeutic applications in thyroid cancer.

Reviewer 2 Report
Comments and Suggestions for Authors
The role of mitochondrial genome stability and metabolic plasticity in thyroid cancer,' the authors provide a comprehensive overview of the role of mitochondrial genome instability and metabolic plasticity in cancer initiation, progression, and management. The review highlights the genomic variability in mitochondrial DNA, metabolic changes, and their crosstalk in thyroid cancer. The review explores emerging evidence regarding mitochondrial genomic instability and metabolic adaptation in thyroid cancer, which is an area of rapidly growing interest and needs focused review on this cancer type. The review not only discusses the mitochondrial genomic variability and metabolic changes in thyroid cancer but also connects the links between both of them. It explores mechanistic interaction between mtDNA instability, OXPHOS alterations, redox biology and metabolic regulatory signaling. The manuscript is well written and provides sufficient information about gaps in the current knowledge and future directions. Table 1 provides a useful resource of known mtDNA alterations in thyroid cancer and the figures are clear and according to the text. The authors have also provided information about gaps in the knowledge, comments about inconsistent findings and diagnostic and therapeutic opportunities. The references are comprehensive and up to date.
Overall, the review is timely and well-written and provides novel insights in mitochondrial genetics and metabolic plasticity of thyroid cancer. The following are the few minor suggestions:
1. Reference is missing for the following text:
Line 344-346 = “Studies have also revealed that certain nanodrugs, such as the
KLA-Au system, can induce mtDNA leakage to activate innate immune responses and potentiate anti-tumor immunity.
2. Figures and Table 1 should be referenced at appropriate places in the main text.
Author Response
Dear Reviewer,
We thank you for the positive assessment of our work. Following your suggestions, the revised manuscript has be polished by a native English speaker and we thoroughly revised the manuscript and addressed all issues. Revisions to the manuscript are marked using the “Track Changes” function. The corrections in the paper and the response are as follows:
Comments 1. Reference is missing for the following text: Line 344-346 = “Studies have also revealed that certain nanodrugs, such as the KLA-Au system, can induce mtDNA leakage to activate innate immune responses and potentiate anti-tumor immunity.
Response: Thank you for pointing this out. We have revised the text to avoid overstatement and added appropriate references. Specifically, we now state: “In addition, nanoparticle-based approaches, such as KLA-Au systems, have been re-ported to selectively accumulate in mitochondria, enhance oxidative damage, and may promote mtDNA release and innate immune activation[97-99]”. This correction provides accurate attribution and ensures that the statement is properly supported by relevant literature.
References are as followings:
- Ma, X.; Wang, X.; Zhou, M.; Fei, H. A mitochondria-targeting gold-peptide nanoassembly for enhanced cancer-cell killing. Adv Healthc Mater 2013, 2, 1638-1643, doi:10.1002/adhm.201300037.
- Kodiha, M.; Wang, Y.M.; Hutter, E.; Maysinger, D.; Stochaj, U. Off to the organelles - killing cancer cells with targeted gold nanoparticles. Theranostics 2015, 5, 357-370, doi:10.7150/thno.10657.
- Xing, Y.; Peng, A.; Yang, J.; Cheng, Z.; Yue, Y.; Liu, F.; Li, F.; Liu, Y.; Liu, Q. Precisely Activating cGAS-STING Pathway with a Novel Peptide-Based Nanoagonist to Potentiate Immune Checkpoint Blockade Cancer Immunotherapy. Adv Sci (Weinh) 2024, 11, e2309583, doi:10.1002/advs.202309583.
Comments 2. Figures and Table 1 should be referenced at appropriate places in the main text.
Response: Thank you for your valuable comment. We have carefully revised the manuscript and ensured that all Figures, Table 1 and Table 2 are appropriately referenced at relevant positions in the main text. Figure 1 is now cited in the section 2.2、2.3、2.4. Table 1 is referenced in the section 2.2(Page 2, Line 80). Table 2 is referenced in the section 5(Page 9, Line 391).

Reviewer 3 Report
Comments and Suggestions for Authors
The manuscript provides a comprehensive and timely review of mitochondrial genome instability and metabolic plasticity in thyroid cancer. However, the manuscript just primarily summarizes existing literature without offering sufficiently new insights, hypotheses, or frameworks. Several recent reviews (2022–2025) have already addressed similar themes, and the current work does not clearly differentiate itself.
Major:
- Much of the discussion is descriptive rather than analytical. Controversial issues, such as the diagnostic relevance of D-loop mutations and inconsistencies in mtDNA copy number findings are mentioned but not critically evaluated. This limits the paper’s impact and scholarly contribution. Maybe, in sections discussing the clinical application of mtDNA (e.g., D-loop mutations, mtDNA copy number), please emphasize the current limitations and controversies more clearly to avoid potential misinterpretation.
- The manuscript frequently highlights the potential of mtDNA as a diagnostic or therapeutic target, but without adequate discussion of the technical limitations and translational barriers. This creates a risk of overstating the actual clinical applicability.
-
Please consider condensing some repetitive statements in the Discussion section (e.g., on isolation/purity, storage, and immune rejection), which appear multiple times.
Minor :
-
Please include more recent studies (from the past 3–5 years) to strengthen the discussion and highlight the latest advances in the field. In addition, ensure consistency in reference style (uniform DOI presentation, journal abbreviations, and page numbers).
-
Define key abbreviations (e.g., DAMPs, EVs, OXPHOS) upon first mention for the benefit of non-specialist readers.
-
Revise table titles to more explicitly reflect their content (e.g., add “clinical progress summary” to Table 2).
Author Response
Dear reviewer,
We are highly obliged for your positive assessment of our work. We have thoroughly revised the manuscript and addressed the issues. Revisions to the manuscript has been marked up using the “Track Changes” function. The corrections in the paper and the response are as follows:
Major:
Comments 1. Much of the discussion is descriptive rather than analytical. Controversial issues, such as the diagnostic relevance of D-loop mutations and inconsistencies in mtDNA copy number findings are mentioned but not critically evaluated. This limits the paper’s impact and scholarly contribution.
Response: We sincerely thank you for this insightful suggestion. The manuscript has been revised to strengthen analytical interpretation, highlighting mechanistic insights, methodological considerations, and biological context, while reducing purely descriptive statements. We have expanded our discussion to provide a more critical evaluation of controversial findings:
D-loop mutations: We now highlight that most D-loop variants in thyroid tumors exhibit heteroplasmy rather than homoplasmy, suggesting that only a subset may reach a functional threshold to influence tumor biology. Furthermore, the elevated mutation rate in this region may represent a secondary consequence of mtDNA replication stress or deletions, rather than a direct driver event. These considerations raise important questions regarding the actual diagnostic relevance of D-loop mutations and underscore the need for functional validation. (Page 3, Section 2.2, line 107-114 and 121-134)
mtDNA copy number variations: We have added discussion on the discrepancies between tissue-based and blood-based studies, where copy number is frequently elevated in tumor tissues but decreased in peripheral blood. We also note that large-scale genetic studies have failed to establish a causal link between mtDNA copy number and thyroid cancer risk. These inconsistencies may reflect tumor subtype heterogeneity, methodological variability, or confounding clinical factors. We emphasize that the biological significance of copy number alterations remains uncertain and requires further clarification. (Page 4, Section 2.4, line 170-176)
By incorporating these points, we aimed to strengthen the analytical depth of the discussion and provide readers with a more balanced understanding of the unresolved questions in this field.
Maybe, in sections discussing the clinical application of mtDNA (e.g., D-loop mutations, mtDNA copy number), please emphasize the current limitations and controversies more clearly to avoid potential misinterpretation.
Response: Thank you for this insightful suggestion. We agree that a clear emphasis on limitations is crucial for interpreting the potential clinical applications of mtDNA research.
In response, we have revised the manuscript to more explicitly outline the current controversies and limitations. Specifically, in the clinical application sections, we have now more strongly emphasized the conflicting evidence regarding D-loop mutations and highlighted the current lack of robust studies establishing the sensitivity and specificity of mtDNA copy number as a clinical tool (Page 8, Section 5, line 347-355).
We believe these revisions provide a more balanced and critical perspective, helping to guide the reader's interpretation and prevent overstatement of the current findings.
Comments 2. The manuscript frequently highlights the potential of mtDNA as a diagnostic or therapeutic target, but without adequate discussion of the technical limitations and translational barriers. This creates a risk of overstating the actual clinical applicability.
Response: Thank you for raising this critical point. We sincerely appreciate your insightful comment regarding the importance of balancing the discussion of potential applications with a clear acknowledgment of existing challenges.
We have now integrated dedicated paragraphs discussing current technical limitations and translational barriers at the conclusion of Sections 2.2, 2.3, 2.4, and 5. These revisions explicitly address key hurdles, such as the delivery challenges for mitochondrial-targeted therapies, the standardization of detection methods for mtDNA biomarkers, and the complexities of achieving clinical-grade specificity.
We believe these additions provide a more realistic and scholarly perspective on the current state of the field, mitigating the risk of overstating immediate clinical applicability and offering readers a more balanced view of the path from basic research to clinical translation.
Comments 3. Please consider condensing some repetitive statements in the Discussion section (e.g., on isolation/purity, storage, and immune rejection), which appear multiple times.
Response: Thank you for this valuable observation. We have carefully reviewed the entire manuscript, particularly the Discussion section, to identify and condense repetitive statements concerning isolation/purity, storage, and immune rejection. Redundant passages have been removed or consolidated to streamline the narrative and improve the overall conciseness of the text. We believe these revisions have significantly enhanced the clarity and focus of the discussion.
Minor:
Comments 1. Please include more recent studies (from the past 3–5 years) to strengthen the discussion and highlight the latest advances in the field. In addition, ensure consistency in reference style (uniform DOI presentation, journal abbreviations, and page numbers).
Response: We appreciate your constructive feedback. In response, we have updated the manuscript to incorporate recent studies from the past 3–5 years, ensuring that the discussion reflects the latest advances in the field. Additionally, we have standardized the reference formatting to ensure consistency in DOI presentation, journal abbreviations, and page numbers throughout the manuscript.
We have updated the clinical applications section by including recent studies on mitochondria-targeted therapeutic strategies, such as metabolic inhibitors, ROS modulators, and novel mitochondrial genome editing approaches [88,91-96,99,101-103]. Additionally, we have incorporated recent research findings in the discussion of current limitations, highlighting challenges related to sensitivity, specificity, and translational feasibility [40, 58, 64, 66, 72-73].
Comments 2. Define key abbreviations (e.g., DAMPs, EVs, OXPHOS) upon first mention for the benefit of non-specialist readers.
Response: Thank you for this helpful suggestion. We have now carefully reviewed the manuscript and defined all key abbreviations—including DAMPs (damage-associated molecular patterns), EVs (extracellular vesicles), and OXPHOS (oxidative phosphorylation)—upon their first occurrence in the text. We believe this revision will improve the accessibility of the content for readers across different subfields.
We appreciate your thoughtful comment aimed at enhancing the clarity of our manuscript.
Comments 3. Revise table titles to more explicitly reflect their content (e.g., add “clinical progress summary” to Table 2).
Response: Thank you for pointing this out. We have taken your feedback into consideration and, as the initial draft did not originally contain a Table 2, we have now added a new table summarizing the clinical progress of key mitochondrial biomarkers and therapies. This table has been explicitly titled "Table 2. Clinical Applications of mtDNA in Thyroid Cancer Diagnosis and Therapy." to clearly reflect its content. We believe this addition enhances the clarity and usefulness of the manuscript for readers.

Round 2
Reviewer 3 Report
Comments and Suggestions for Authors
The revised manuscript has satisfactorily addressed all previous comments. The authors have substantially improved the analytical depth and provided balanced discussion on controversial aspects of mtDNA alterations. The technical and translational limitations are clearly described, and repetitive content has been removed. Only a minor suggestion would be to include one or two additional recent mechanistic references regarding D-loop mutations.
Overall, the revision is thorough and the manuscript is suitable for publication in Biomedicines.
Author Response
Dear Reviewer,
We wish to express our sincere gratitude for the positive and constructive feedback on our manuscript. All the reviewers' comments have been addressed in the revised version, and the corresponding changes have been meticulously marked using the "Track Changes" function.
Comments 1. Only a minor suggestion would be to include one or two additional recent mechanistic references regarding D-loop mutations..
Response: We are grateful for this insightful comment. In response, we have incorporated a discussion of recent mechanistic studies that elucidate the role of D-loop mutations. The added text (Page 6, lines 269-277) describes pan-cancer analyses indicating that somatic D-loop mutations are influenced by tumor-specific evolutionary selection, alongside evidence that mutations in non-HVS regions and chemotherapy-induced mutations in HVS1 are associated with reduced mtDNA copy number and changes in chemosensitivity [61,62].
References are as followings:
- Ji, X.; Guo, W.; Gu, X.; Guo, S.; Zhou, K.; Su, L.; Yuan, Q.; Liu, Y.; Guo, X.; Huang, Q.; et al. Mutational profiling of mtDNA control region reveals tumor-specific evolutionary selection involved in mitochondrial dysfunction. EBioMedicine 2022, 80, 104058, doi:10.1016/j.ebiom.2022.104058.
- Harino, T.; Tanaka, K.; Motooka, D.; Masuike, Y.; Takahashi, T.; Yamashita, K.; Saito, T.; Yamamoto, K.; Makino, T.; Kurokawa, Y.; et al. D-loop mutations in mitochondrial DNA are a risk factor for chemotherapy resistance in esophageal cancer. Sci Rep 2024, 14, 31653, doi:10.1038/s41598-024-80226-3.
